# Methicillin-Resistant *Staphylococcus aureus* Clonal Complex 398 as a Major MRSA Lineage in Dogs and Cats in Thailand

**DOI:** 10.3390/antibiotics10030243

**Published:** 2021-02-28

**Authors:** Surawit Chueahiran, Jitrapa Yindee, Pongthai Boonkham, Nipattra Suanpairintr, Pattrarat Chanchaithong

**Affiliations:** 1Department of Veterinary Microbiology, Faculty of Veterinary Science, Chulalongkorn University, Bangkok 10330, Thailand; 6271026031@student.chula.ac.th (S.C.); jitrapa.y@chula.ac.th (J.Y.); 2Veterinary Diagnostic Laboratory, Faculty of Veterinary Science, Chulalongkorn University, Bangkok 10330, Thailand; pongthat.b@chula.ac.th; 3Department of Pharmacology, Faculty of Veterinary Science, Chulalongkorn University, Bangkok 10330, Thailand; nipattra.d@chula.ac.th; 4Research Unit in Microbial Food Safety and Antimicrobial Resistance, Faculty of Veterinary Science, Chulalongkorn University, Bangkok 10330, Thailand

**Keywords:** MRSA, clonal complex 398, antimicrobial resistance, cat, dog

## Abstract

The aim of this study was to present molecular and antimicrobial resistance characteristics of methicillin-resistant *Staphylococcus aureus* (MRSA) clonal complex (CC) 398 isolated from diseased dogs and cats in Thailand. A total of 20 MRSA isolates of 134 *Staphylococcus aureus* isolated from canine and feline clinical samples during 2017–2020 were CC398, consisting of sequence type (ST) 398 (18 isolates), ST5926 (1 isolate), and ST6563 (1 isolate) by multilocus sequence typing. *spa* t034 and staphylococcal cassette chromosome *mec* (SCC*mec*) V were predominantly associated with ST398. Intraclonal differentiation was present by additional *spa* (t1255, t4653), non-detectable *spa*, composite SCC*mec* with a hybrid of *ccrA1B1*+*ccrC* and class A *mec* complex, and DNA fingerprints by pulsed-field gel electrophoresis. The isolates essentially carried antimicrobial resistance genes, mediating multiple resistance to β-lactams (*mecA*, *blaZ*), tetracyclines [*tet*(M)], aminoglycosides [*aac(6′)-Ie-aph(2′)-Ia*], and trimethoprim (*dfr*). Livestock-associated MRSA ST398 resistance genes including *lnu*(B), *lsa*(E), *spw*, *fexA*, and *tet*(L) were heterogeneously found and lost in subpopulation, with the absence or presence of additional *erm*(A), *erm*(B), and *ileS2* genes that corresponded to resistance phenotypes. As only a single CC398 was detected with the presence of intraclonal variation, CC398 seems to be the successful MRSA clone colonizing in small animals as a pet-associated MRSA in Thailand.

## 1. Introduction

Methicillin-resistant *Staphylococcus aureus* (MRSA) is a multidrug-resistant bacterium widespread in human and domestic animals, and is associated with a wide spectrum of infection, from superficial to invasive and systemic infections [1]. Of medical importance, nosocomial infections are caused by a particular subpopulation, known as hospital-associated MRSA (HA-MRSA). Molecular epidemiological tools, such as multilocus sequence typing (MLST), reveal HA-MRSA clonal lineages in terms of clonal complex (CC) and sequence types (ST), including CC8 (ST239), CC22 (ST22), CC30 (ST26) [2]. In addition, CC8 (ST8), CC80 (ST80), CC30 (ST80), CC59 (ST50), and CC93 (ST93) are classified as community-associated MRSA (CA-MRSA) [3]. The nosocomial spread of CA-MRSA in healthcare settings has been reported in many circumstances [4,5,6]. MRSA-colonizing cattle, pigs and poultry are clonal-specific, for CC398 (ST398), CC9 (ST9), CC5 (ST5), and CC49 (ST49), among others, which are all known as livestock-associated MRSA (LA-MRSA). Various MRSA clones have been identified in dogs, cats and horses. Human-related MRSA clones are a major population in small companion animals, and CC398 has been sporadically identified as a subpopulation [7,8,9].

*S. aureus* may colonize several animal host species, including dogs and cats, but *S. pseudintermedius* and *S. schleiferi* subsp. *coagulans* predominate in canine healthy carriers and in infections. Clonal expansion and multidrug resistance of methicillin-resistant *S. pseudintermedius* (MRSP) limit effective antimicrobial treatment in small animal practice [10]. Likewise, MRSA is occasionally isolated from canine and feline clinical samples, particularly from skin and soft tissue infections [1,11]. Risk factors contributing to infection in pets include antimicrobial exposures, hospital stays and medical implants [12]. Furthermore, small companion animals can play the role of MRSA reservoirs in the community, and can be a source of infection in humans [13,14]. Interspecies transmission among risk groups including veterinary staff, pet owners and healthy pet carriers was proven using bacterial genetic identities [15]. Recently, increasing reports describing MRSA infections in animals and animal-to-human transmission have raised the alarm of a possible emerging multidrug-resistant pathogen [16,17]. Previous studies illustrate that various human- and livestock-related MRSA clones have been recovered from canine and feline patients and carriers [7,8,9,15,18,19]. Little is known about the clonal spread of MRSA, which is associated with diseases in dogs and cats in Thailand. Genetic features of canine and swine MRSA ST398 from animal carriers were the first-characterized, and have both common and different traits [20,21]. From 2017 to 2020, MRSA were sporadically isolated from canine and feline clinical samples from our routine diagnostic activity, and the majority of the isolates shared some common resistance phenotypes, such as resistance to clindamycin (lincosamides) and susceptibility to erythromycin (macrolides), which are similar to the canine MRSA ST398 isolate in the previous study [20]. MLST analysis illustrated that all clinical MRSA isolates were clonally related to CC398, and other MRSA clones were not detected. This study presents MRSA CC398 isolation from canine and feline clinical samples, along with its intraclonal variation by genetic and antimicrobial resistance features.

## 2. Results

### 2.1. Numbers and Origins of MRSA

Of 7499 canine and feline clinical samples, 134 *S. aureus* was primarily identified during 2017–2020 by Vitek®2 automated identification (ID). Twenty *S. aureus* isolates exhibited positive MRSA during screening by cefoxitin via Vitek®2 antimicrobial susceptibility testing (AST), and resistance to cefoxitin via disk diffusion method. All cefoxitin-resistant isolates were *mecA*-positive by PCR. MRSA was recovered from submitted samples each year in 7 of 1772 samples (2017), 5 of 1864 samples (2018), 5 of 2690 samples (2019), and 3 of 1173 samples (January to June 2020). The MRSA-positive samples were submitted from three veterinary hospitals in Bangkok, Thailand. Of 20 MRSA isolates, 15 and 5 isolates were recovered from canine and feline samples, respectively. Lesions or diseases associated with MRSA infection were wounds and abscesses (16), surgical site infections (3), and urinary tract infections (1).

### 2.2. Genetic Characteristics of Canine and Feline MRSA CC398

From MLST analysis, eighteen MRSA isolates were ST398 (allelic profile 2-35-19-2-20-26-39). Two isolates were ST5926 and a new ST6563, each of which are single locus variants (SLV) of ST398 by *tpi* gene. The new ST6563 had a new allele variation (allelic number 740) of *tpi* (A300T). Using an 80% cutoff for similarity, DNA fingerprint patterns by pulsed-field gel electrophoresis (PFGE) of the isolates were clustered into 3 pulsotypes (Figure 1). Fourteen and one MRSA ST398 isolates contained *spa* t034 and t4652, respectively. *spa* t1255 was associated with ST5926 and ST6563. Three MRSA ST398 were *spa*-negative by PCR, and had a composite staphylococcal cassette chromosome *mec* (SCC*mec*) which consisted of a class C *mec* complex and two *ccr* complexes, including *ccrA1B1* and *ccrC*. This composite SCC*mec* was also found in one MRSA-ST398-t034 isolate; however, SCC*mec* V was predominantly associated with 14 ST398 isolates, ST5956 and ST6563. 

By broth microdilution, all MRSA were multidrug-resistant (more than 3 antimicrobial classes) and exhibited resistance to, at least, penicillin and cefoxitin (β-lactams), tetracycline (tetracyclines), clindamycin (lincosamides) and ciprofloxacin (fluoroquinolones). No resistance to vancomycin, linezolid, rifampicin and fusidic acid was found in all isolates. Resistance to antimicrobials in high proportions was observed for trimethoprim (19/20), kanamycin (18/20), gentamicin (15/20), and tiamulin (16/20). Distribution of the isolates into categories of susceptible, intermediate or resistant to each antimicrobial agent, along with minimum inhibitory concentrations (MIC), are illustrated in Table 1. The resistance rate of erythromycin, chloramphenicol, and mupirocin was 4/20, 5/20, and 4/20, respectively. 

### 2.3. Antimicrobial Resistance Gene Carriage

Only *mecA* and *tet*(M) were detected in all MRSA isolates. *bla*Z and *tet*(L) were contained in 18/20 and 16/20 isolates, respectively. Aminoglycoside or aminocyclitol modifying enzyme (AME)-encoding genes were present, including *ant(4′)-Ia* (17/20), *ant(6)-Ia* (16/20), bifunctional *aac(6′)-Ie-aph(2′)-Ia* (18/20), and *spw* (16/20). Only one isolate did not contain AME-encoding genes. The sixteen *spw*-positive isolates also contained *lnu*(B) and *lsa*(E), and presented a higher MIC for tiamulin (>4 μg/mL). Nineteen of 20 isolates carried either *dfrA* (16/20) or *dfrG* (3/20), and one isolate had both *dfr*A and *dfrG*. The four *dfrG*-positive isolates contained *erm*(B). Both *erm*(A) and *erm*(B) were found in the same three isolates, and one isolate carried only *erm*(B). *fexA* was detected in five chloramphenicol-resistant isolates, but *cat*_pC221_ was not found. Four isolates presenting high-level mupirocin resistance contained *ileS2*. The antimicrobial resistance gene profiles are shown in Table 2.

### 2.4. Intraclonal Variation and Common Features of Canine and Feline MRSA CC398

Thirteen MRSA isolates were ST398-V-t034 (ST-SCC*mec*-*spa*), which was a major characteristic in both pulsotype A and B, and contained common antimicrobial resistance genes, including *mecA*, *ant(6)-Ia*, *aac(6′)-Ie-aph(2′)-Ia*, *tet*(M) and *tet*(L), single *dfrA*, *lnu*(B), *lsa*(B), and *spw*. The ST398-V-t034 pulsotype B were mostly found in 2017, whereas the pulsotype A was found in 2018–2020 (Figure 1). The composite SCC*mec* was associated with three *spa*-negative ST398 isolates presenting >90% DNA fingerprint similarity, and one ST398-t034 isolate, which was grouped in pulsotype A and recovered during 2018–2020 (Figure 1). The common features of ST398 containing the composite SCC*mec* were non *ant(6)-Ia, lnu*(B), *lsa*(B) and *spw* carriage, and a positive result for *erm*(B) and *dfrG* detection. The ST5926-V-t1255 and ST6563-V-t1255 isolates present an identical DNA fingerprint pattern (Figure 1). One MRSA-ST398-V-t4562 pulsotype B was isolated in 2017.

## 3. Discussion

This study presents MRSA CC398 isolated of canine and feline origin which were associated with diseases. With no other CC detected, the evidence shows that CC398 is the major MRSA lineage in cats and dogs in Thailand. MRSA ST398 was first detected in a Thai dog carrier in 2010 [20]. Despite a low prevalence of MRSA in the clinical samples, ST398 and their SLVs detected in this study represent opportunistic infection of this clone in small companion animals. MRSA ST398 was carried by veterinary practitioners, and was also the dominant LA-MRSA clone in pigs and swine workers in Thailand [20,21]. It was the most significant MRSA clone associated with animals and animal-related occupations. CC398 have been heterogeneously isolated together with other sequence types from small animals in other countries, but the majority of the isolates are human-related MRSA clones, such as CC5, CC8, CC22, CC45, and CC239 [8,9,11,15,22]. MRSA CC398 was found in 21.8% and 17.6% of canine and feline MRSA isolates over a five-year period (2010–2015) in France, respectively [8], and was not present in some countries [13,15,19]. In addition, there is no information regarding human-related MRSA clones in companion animals in Thailand. Since its first detection in 2010, we speculate that this clone has persisted and spread, and was occasionally associated with diseases in canine and feline hosts. 

ST398-V-t034 pulsotype B was the major population, and was mostly isolated in 2017. ST398-V-t034 is sporadically found in canine carriers in China [22] and in diseased dogs and cats in Austria [9]. The major *spa* t011 and SCC*mec* IV are predominant in equine, canine and feline MRSA ST398 in France and Austria [8,9]. t034 is the most prevalent *spa* associated with CC398 in Asian countries [20,22,23]. In addition to a single-nucleotide polymorphism, SLV ST5926 and ST6563 had a different *spa* t1255, which represented a minor variation mirroring evolution of the CC398 in cats and dogs. SCC*mec* IV and V were more common in ST398 isolated from companion animals [8,9], while only SCC*mec* V is more abundant in LA-MRSA ST398 [21,24,25]. The type V and the composite SCC*mec* with a hybrid of type 1 *ccr* and *ccrC* is also contained in swine MRSA ST398 in Thailand [21]. The Thai swine strains with the composite SCC*mec* contained *spa* t034, but the *spa* could not be detected in the majority of canine and feline ST398-composite SCC*mec* strains, together with the loss of genes presenting in LA-MRSA. The SCC*mec* and *spa* diversifications may be a result of the plasticity of *orfX*-surrounding regions where SCC*mec* and *spa* are located, which promote *S. aureus* genetic events [26]. However, the presence or absence of *spa* and the variation of the composite SCC*mec* need to be deduced by further whole genome analysis. These common and minor different traits may reflect a concomitant CC398 ancestor in a variety of animals, as well as bacterial divergent evolution in different hosts in Thailand. 

Antimicrobial resistance genes carried by the present canine and feline MRSA CC398 population were heterogenous. In addition to *mecA*, multidrug resistance was mediated by common resistance genes in staphylococcal species, such as *blaZ, tet(M), ant(4′)-Ia*, *ant(6)-Ia*, and *aac(6′)-Ie-aph(2)-Ia* [27,28]. *tet*(M) is frequently distributed in ST398 derived from companion animals, and it can be a differentiation marker from human-related clones [8]. In Thailand, *dfrA* is remarkably found in ST398-V-t034 from small animals, but LA-MRSA from pigs and MRSP contains *dfrG* [20,21]. The antimicrobial resistance gene carriages of Thai ST398-V-t034 from swine and from small animals were different. *erm* genes were not found in ST398-V-t034, in contrast to a wide distribution of *erm*(A) and *erm*(C) found in the strains from pigs [21]. However, *dfrG* was detected in the ST398-composite SCC*mec* subpopulation with the presence of *erm*(B) genes, similar to the resistance gene profiles of canine MRSP [27,29]. Taken together, the loss of LA-MRSA-associated resistance genes, including *lnu*(B), *lsa*(E), *spw*, *tet*(L), and *fexA* [30,31,32], likely corresponds to antimicrobial use patterns in small-animal practice, which are different than in livestock. In addition, the presence of *ileS2* possessing high-level mupirocin resistance may be due to pressure from the introduction of topical mupirocin in small-animal veterinary practice [33]. The heterogenous resistance in the canine and feline MRSA CC398 populations indirectly presented a picture of resistance gene loss and gain in this clone to be fit in canine and feline niches. 

The isolates exhibited a resistance to multiple antimicrobial classes that limited the first-line and second-line therapeutic options in small animal practices, such as β-lactams, gentamicin, clindamycin, doxycycline and sulfamethoxazole/trimethoprim. Topical treatment may be an alternative method for wound and skin infection in the most common cases [34,35]. However, systemic treatment and extraordinary use of last-resort antimicrobials is required in the case of invasive infections. Additional resistance can be developed by introduction of new drugs in veterinary practice. Florfenicol and rifampicin treatment were associated with the presence of *fexA*, *cfr,* and *rpoB* mutation in MRSP, respectively [36,37]. Indeed, antimicrobial stewardship programs by effective diagnostic approaches, antimicrobial resistance monitoring program and judicious antimicrobial use are strongly recommended to follow up and diminish ongoing development of resistance.

MRSA was recovered from only 0.26% of clinical samples from dogs and cats. The prevalence of MRSA in companion animals is <2% [7,22]. The proportion of methicillin resistance in *S. aureus* isolates was 20/134 (14.92%), which was a range similar to a previous study [19], but which cannot be completely compared to other studies, due to the sample sources and population. The majority of the isolates were associated with wound infections and abscess. A wide-scale study in Germany reported that 62.7% of canine wounds were infected by MRSA [11]. Even with low prevalence in companion animals, attention should be continuously paid for *S. aureus* due to the wide species distribution and their adaptive ability in different hosts, especially MRSA CC398. Furthermore, many studies demonstrated that veterinary staff share common MRSA strains with companion animals and small-animal hospital environments [16,38,39], owing to the proximity and frequent contact. Monitoring at-risk groups, including veterinary personnel and pet owners, healthy pet carriers in the community and environmental contamination should be undertaken in the term of one health approach. 

Clinical MRSA isolates in the present study manifested pathogenic potential of CC398 in dogs and cats. *S. aureus* is more frequently recovered from cat wounds; however, the proportion of MRSA from dog wounds is higher [11]. Circulation of MRSA ST398 in dog kennels can be a source of a variety of infections, including gangrenous mastitis in bitches, and pyoderma and fatal pneumonia in puppies [40]. The evidence from this study supported MRSA ST398 as being not only a recognized worldwide LA-MRSA, but also a pet-associated MRSA. We are unable to speculate on the replacement of previous MRSA clones because prior epidemiological information is lacking. Nevertheless, CC398 has successfully expanded in horses, replacing the previous predominant equine CC8 in Europe [8,9]. Genetic changes supporting clonal proliferation and mechanisms for host adaptation of *S. aureus* in canine and feline hosts are likely, supported by information obtained from other domestic animals [41,42,43]. Genes encoding survival and virulence factors were not examined in our study, and need to be explored. Little is known about the genetic adaption of *S. aureus* colonization in dogs and cats. Immune evasion cluster (IEC) delivered by bacteriophages, a factor enhancing MRSA colonization in humans, is inconsistently found in ST398 strains from companion animals, but completely absent in LA-MRSA ST398 [18,41]. Host-specific virulence factors or genome adaptations are not identified in ST22 (EMRSA-15) from humans or companion animals in the UK [44]. High-resolution genome analysis should be further investigated for understanding the complete evolutionary relationship of this clone among domestic animals. Specific markers from deep insights could be informative in term of genomic surveillance for tracing cross-species transmission and for developing preventive measures in the companion animal and livestock sectors. 

## 4. Materials and Methods

### 4.1. Bacterial Isolates

*Staphylococcus aureus* isolates were obtained from clinical samples, from diseased cats and dogs submitted to the Veterinary Diagnosis Laboratory of Chulalongkorn University, Bangkok, Thailand, from January 2017 to June 2020. The species identification and antimicrobial susceptibility testing was routinely performed via the Vitek® automated system, with ID-GP and AST-GP81 cards (bioMérieux, France) for clinical purposes. Bacterial strains were stocked in an −80 °C freezer with 20% glycerol in tryptic soy broth. The *S. aureus* isolates presenting cefoxitin resistance by Vitek® were included for species confirmation, by Microflex Biotyper matrix-assisted laser desorption/ionization-time of flight mass spectrometry (MALDI-TOF MS) (Bruker Dalnotics, Germany) and *nuc* gene PCR [45], after recovery onto 5% sheep blood agar and DNA extraction using Nucleospin Tissue DNA extraction kits (Machery-Nagel, Germany). The DNA was also used in all PCR-based molecular assays.

### 4.2. Antimicrobial Susceptibility Testing and Resistance Gene Detection

Cefoxitin disk diffusion test was performed for phenotypic confirmation of MRSA, and *mecA* was detected by PCR [46]. MIC of antimicrobials was determined by broth microdilution using Sensititre customized EUST 96-well plates (Trek Diagnostic Systems, United Kingdom). Resistance, intermediate and susceptible results were justified by clinical breakpoint from the Clinical and Laboratory Standard Institute, as follows: penicillin, cefoxitin, gentamicin, kanamycin, streptomycin, tetracycline, erythromycin, clindamycin, ciprofloxacin, chloramphenicol, sulfamethoxazole, trimethoprim, vancomycin, rifampicin and linezolid [47,48]. Despite no CLSI-provided breakpoint for mupirocin, fusidic acid, and tiamulin, interpretation from scientific articles was referred to [49,50,51].

Acquired antimicrobial resistance genes encoding mechanisms of resistance to various antimicrobials were detected via PCR, including *blaZ* (β-lactamase), *tet*(M) (ribosomal protective proteins), *tet*(K) and *tet*(L) (tetracycline efflux proteins), *ant(4′)-Ia*, *ant(6)-Ia* and bifunctional *aac(6′)-Ie-aph(2′)-Ia* (aminoglycoside modifying enzymes), *dfrA* and *dfrG* (dihydrofolate reductase inhibitors), *erm*(A), *erm*(B) and *erm*(C) (erythromycin resistance methylase), *lnu*(B) (lincosamide nucleotydyl transferase), *lsa*(E) (lincosamide-streptogramin A efflux protein), *fexA* (florfenicol exporter), *cat*_pC221_ (chloramphenicol acetyltransferase) and *ileS2* (alternative isoleucyl transferase). Specific primers are listed in Appendix A. All PCR were performed in a total of 25 μL solution, using 5X Firepol Master Mix (Solis Biodyne, Estonia) with 0.2 pmol/μL of each primer. The PCR conditions included an initial denaturation at 95 °C for 3 min, 30 cycles of denaturation at 95 °C for 30 s, annealing at the specific temperature for each primer for 30 s and extension at 72 °C for the specific duration depending on the amplicon size (Appendix A), followed by a final extension at 72 °C for 5 min.

### 4.3. DNA Fingerprint Analysis

DNA fingerprints of the MRSA isolates were illustrated by *Cfr*9I macro-restriction and PFGE [52]. Bacterial cells in agarose plugs were lysed by lysozyme, lysostaphin and detergents. After the plug-washing steps, chromosomal DNA was digested in 50 U of *Cfr*9I (Thermo Fisher Scientific, CA, USA). Macro-restriction fragments were separated in 1% agarose gel and TE buffer, with a switch time of 5–40 sec and 6 V/cm for 21 h, using CHEF-DRIII PFGE (Bio-Rad, CA, USA). The DNA band patterns were stained with ethidium bromide and documented under a UV illuminator. Relationships among strains were analysed by dendrogram construction using Dice coefficients, using the unweighted matrix pair group method with arithmetic mean (UPGMA) by Bionumeric software version 7.6 (Applied Maths, Belgium) with 1.5% optimization and 1.5% position tolerance. Pulsotypes were classified at >80% band pattern similarity, representing close clonal relatedness. 

### 4.4. spa Typing

Polymorphic X region of *spa* gene was amplified by 1095F (5′-AGACGATCCTTCGGTGAGC-3′) and 1517R (5′-GCTTTTGCAATGTCATTTACTG-3′) primers [53]. The strains which resulted in negative amplification by 1095F and 1517R were included for *spa* PCR by additional primers, [spa-1113f (5′-TAA AGA CGA TCC TTC GGT GAG C-3´) and spa-1514r (5′-CAG CAG TAG TGC CGT TTG CTT -3´)] [54]. Amplicons were purified using a Nucleospin DNA purification kit (Machery-Nagel, Germany) for capillary Sanger’s DNA sequencing at a service company (Apical Scientific Co., Ltd., Malaysia). *spa* sequences were analysed by comparing to *spa* tandem repeats in the database (http://spaserver.ridom.de/ (accessed on 3 February 2021)).

### 4.5. SCCmec Identification and MLST

SCC*mec* types were analysed based on *ccr* complex type and class of *mec* gene complex by multiplex PCRs [55] and interpreted according to criteria [56]. Composite SCC*mec* was classified in cases where more than one *ccr* complex was detected. MLST of *S. aureus* was conducted by amplification and nucleotide sequence analysis of *arcC*, *aroE*, *glpF*, *gmk*, *pta*, *tpi* and *yqiL* [57]. DNA sequences and allelic profiles were electronically submitted to the database (www.pubmlst.org (accessed on 3 February 2021)) for allelic number and ST assignation, respectively.

## 5. Conclusions

CC398 is the major MRSA lineage in dogs and cats in Thailand. By genetic and antimicrobial resistance features, ST398-V-t034 has proliferated in small animals since its first detection in a dog carrier and veterinarians. Minor differentiation between *spa* and SCC*mec* variants and antimicrobial resistance gene carriages described intraclonal variation and its continuous evolution. With only the single clone present in this study, we propose that the MRSA CC398 was a predominant pet-associated MRSA clone of veterinary importance.

## Figures and Tables

**Figure 1 antibiotics-10-00243-f001:**
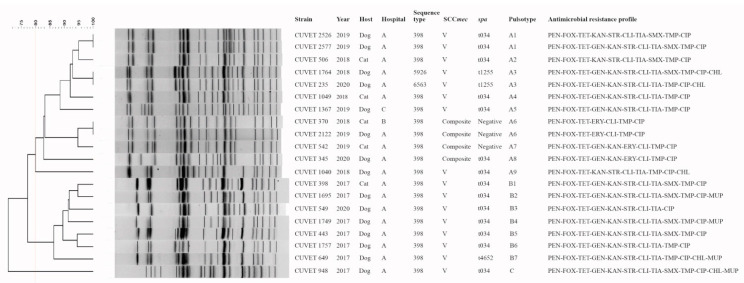
Genetic relatedness and antimicrobial resistance profile of canine and feline methicillin-resistant *Staphylococcus aureus*. PEN, penicillin; FOX, cefoxitin; TET, tetracycline; GEN, gentamicin; KAN, kanamycin; STR streptomycin; ERY, erythromycin; CLI, clindamycin; TIA, tiamulin; SMX, sulfamethoxazole; TMP, trimethoprim; CIP, ciprofloxacin; CHL, chloramphenicol; MUP, mupirocin.

**Table 1 antibiotics-10-00243-t001:** Isolates of methicillin-resistant *Staphylococcus aureus* clonal complex 398 in each minimum inhibitory concentrations of antimicrobials and with antimicrobial resistance genes which corresponded to resistance phenotypes.

Antimicrobials *	Resistance Break Point (μg/mL)	Range Tested (μg/mL)	Number of Isolates in Each MIC (μg/mL) **																Number of Resistant Isolates	Resistance Gene(s) ***	Number of Isolates with the Presence of Resistance Genes ***
			≤0.015	0.003	0.006	0.012	0.25	0.5	1	2	4	8	16	32	64	128	256	512			
PEN	≥0.25	0.0012–2							2 (=)	18 (>)									20/20	*blaZ*; *mecA*	18; 20
FOX	≥1	0.5–16									3 (=)	6 (=)	11 (>)						20/20	*mecA*	20
TET	≥1	0.5–16											20 (>)						20/20	*tet*(M); *tet*(L); *tet*(K)	20; 16; 0
GEN	≥16	1–16							2 (≤)			3 (=)	8 (=)	7 (>)					15/20	*aac(6′)-Ie-aph(2′)-Ia*	18
KAN	≥16	4–64									2 (≤)				9 (=); 9 (>)				18/20	*aac(6′)-Ie-aph(2′)-Ia; ant(4′)-Ia*	18; 17
STR	≥16	4–32									1 (≤)	3 (=)		16 (>)					16/20	*ant(4′)-Ia; ant(6)-Ia*	17; 16
ERY	≥8	0.25–8					4 (≤)	10 (=)	1 (=)	1 (=)		4 (>)							4/20	*erm*(B); *erm*(A); *erm*(C)	4; 3; 0
CLI	≥4	0.12–4									20 (>)								20/20	*lnu*(B); *erm*(B); *erm*(A); *erm*(C)	16; 4; 3; 0
CHL	≥32	4–64									6 (≤)	5 (=)	4 (=)		5 (=)				5/20	*fexA*; *cat*_pC221_	5; 0
SMX	≥512	64–512													6 (≤)	5 (=)		7 (>); 2 (=)	9/20	ND	ND
TMP	≥16	2–32								1 (≤)				19 (>)					19/20	*dfrA*; *dfrG*	17; 4
MUP	≥512	0.5–2 & 256					16 (≤)										4 (>)		4/20	*ileS2*	4
VAN	≥16	1–16							19 (≤)	2 (=)									0/20	ND	0
CIP	≥4	0.25–8										1 (=); 19 (>)							20/20	ND	ND
RIF	≥4	0.015–0.5	19 (≤)			1 (=)													0/20	ND	ND
LZD	≥8	1–8						1 (≤)	5 (=)	13 (=)	1 (=)								0/20	ND	ND
FUS	≥2	0.5–4					20 (≤)												0/20	ND	ND
TIA	>4	0.5–4					2 (≤)	2 (=)			16 (>)								16/20	*lsa*(E)	16

* Abbreviation of antimicrobials: PEN, penicillin; FOX, cefoxitin; TET, tetracycline; GEN, gentamicin; KAN, kanamycin; STR, streptomycin; ERY, erythromycin; CLI, clindamycin; CHL, chloramphenicol; SMX, sulfamethoxazole; TMP, trimethoprim; MUP, mupirocin; VAN, vancomycin; CIP, ciprofloxacin; RIF, rifampicin; LZD, linezolid; FUS, fusidic acid; TIA, tiamulin. ** < in parentheses, less than or equal to the MIC; = in parentheses, equal to the MIC; > in parentheses, more than the MIC; Dark gray shade, level of concentration more than interpretive breakpoints referring resistance; light gray shade, level of concentration of intermediate resistance. *** ND, not determined.

**Table 2 antibiotics-10-00243-t002:** Characteristics and antimicrobial resistance gene profiles of 20 methicillin-resistant *Staphylococcus aureus* clonal complex 398 isolates, including sequence type (ST) 398, ST5926 and ST6563, from canine and feline clinical samples, during 2017–2020.

Sequence Type	SCC*mec*	*spa* Type	Pulsotype	No.	Year (No.)	Host (No.)	Antimicrobial Resistance Genes *															
							*mecA*	*blaZ*	*tet*(L)	*tet*(M)	*ant(4′)-Ia*	*ant(6)-Ia*	*aac(6′)-Ie-aph(2)-Ia*	*dfrA*	*dfrG*	*erm*(A)	*erm*(B)	*lnu*(B)	*lsa*(E)	*spw*	*fexA*	*ileS2*
398	V	t034	B	4	2017 (3) 2020 (1)	Dog (3) Cat (1)																
398	V	t034	B	2	2017 (2)	Dog (2)																
398	V	t4652	B	1	2017 (1)	Dog (1)																
398	V	t034	C	1	2017 (1)	Dog (1)																
398	V	t034	A	2	2018 (2)	Cat (2)																
398	V	t034	A	1	2018 (1)	Dog (1)																
398	V	t034	A	1	2019 (1)	Dog (1)																
398	V	t034	A	2	2019 (2)	Dog (2)																
5926	V	t1255	A	1	2018 (1)	Dog (1)																
6563	V	t1255	A	1	2020 (1)	Dog (1)																
398	Composite	t034	A	1	2020 (1)	Dog (1)																
398	Composite	-	A	1	2019 (1)	Cat (1)																
398	Composite	-	A	1	2018 (1)	Cat (1)																
398	Composite	-	A	1	2019 (1)	Dog (1)																

* gray shade, positive.

## Data Availability

Not applicable.

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
