# Peer review of "Methicillin-Resistant Staphylococcus aureus Clonal Complex 398 as a Major MRSA Lineage in Dogs and Cats in Thailand"

_antibiotics, 2021, doi:10.3390/antibiotics10030243_

Round 1

Reviewer 1 Report

The Authors submitted an article regarding molecular and antimicrobial resistance characteristics of methicillin-resistant Staphylococcus aureus (MRSA) isolated from dogs and cats in Thailand. The work is interesting. However, the manuscript is poorly written and requires extensive editing of the English language. The introduction should provide a clear background for the research topic. However, it reports a lot of information without a connection. Besides, the results are presented in an unclear manner (e.g. Table 1 and 2 are illegible). Therefore, in my opinion, the Authors should reformulate the entire manuscript and resubmit it to the journal.

Reviewer 2 Report

Great job.

Author Response

Response to reviewer 2 comment.

We are thankful to the reviewer for considering our manuscript.

Reviewer 3 Report

Present study presents clonal expansion of MRSA CC398 from canine and feline clinical samples, along with its intra-clonal variation by genetic and antimicrobial resistance features. In general, the MS makes a good impression, the results obtained are methodically and theoretically validated. But I have some comments for improving the MS

  1. Line 68, In the introduction section. Although this study indicates that CC398 is the major MRSA lineage in dogs and cats in Thailand. But the information about using ST398 is not clear enough in the introduction section. I recommend the authors include more explanations about focusing on ST398 marker in the manuscript to strengthen their study and discussion. For example, there are also other markers of CC or ST for aureus, why the authors did not evaluate the other ST?
  2. As for table 1, the authors should explain more about the numbers and brackets within the table to let the readers understand this table easily. There is no such explanation in the table legend.
  3. Line 136. The title for table 2 needs to be revised. Characteristics and antimicrobial resistance gene profiles of 20 methicillin-resistant Staphylococcus aureus “clonal complex in 398 isolates” from canine and feline clinical samples, during 2017–2020”. Clonal complex 398? On the other hand, I recommend the authors revise the title for table 2 in that not only the ST 398 but also the ST 5926 and 6563 are described in table 2. What’s the grey shade within the table? This should be explained in the table legend or footnote.
  4. Line 286-288. The information for PCR procedures seems not be enough. I recommend the authors include more explanations about PCR procedures, only the primer information in the supplementary seems not be enough. The cycle? The denaturing temperature…?

Round 2

Reviewer 1 Report

The Authors implemented my suggestions and I have no further comments.

Author Response

Response to Reviewer 1's comments

We thank to the reviewer to re-evaluate our manuscript again.